# Examining Food Sources and Their Interconnections over Time in Small Island Developing States: A Systematic Scoping Review

**DOI:** 10.3390/nu17142353

**Published:** 2025-07-18

**Authors:** Anna Brugulat-Panés, Cornelia Guell, Nigel Unwin, Clara Martin-Pintado, Viliamu Iese, Eden Augustus, Louise Foley

**Affiliations:** 1MRC Epidemiology Unit, University of Cambridge, Cambridge CB2 0QQ, UK; 2European Centre for Environment & Human Health, University of Exeter Medical School, Penryn TR10 8RD, UK; 3Pacific Centre for Environment and Sustainable Development, The University of the South Pacific, Suva, Fiji; 4School of Agriculture, Food and Ecosystem Sciences, The University of Melbourne, Parkville, VIC 3010, Australia; 5Faculty of Medical Sciences, The University of the West Indies, Bridgetown BB11000, Barbados

**Keywords:** small island developing states, food sources, foodscapes, non-communicable diseases, intergenerational, transition lens, evidence review, classification, mapping

## Abstract

Background: Small Island Developing States (SIDS) face high rates of non-communicable diseases (NCDs), and a key structural driver includes SIDS’ heavy reliance on imported food. Yet, our knowledge about food sources in SIDS is limited. Methods: We systematically searched 14 peer-reviewed databases and 17 grey literature repositories, identifying 56 articles and 96 documents concerning food sources in SIDS. Our study aimed to map these sources while considering broader societal, cultural, and environmental aspects. Results: We found high heterogeneity of food sources beyond store-bought foods, highlighting the complexity of food landscapes in this context. To explore these food sources and their interconnections, we developed a classification including *Aid, Buy, Grow, Share, State and Wild* food sources, and offered contextually-sensitive insights into their variety (types), extent (relevance), nature (characteristics) and changes over time. We developed an interactive open-access evidence map that outlined the identified interconnections between food sources following our proposed classification. There are numerous interrelations between food sources, showing that pathways from food sourcing to consumption can be unexpected and complex. Conclusions: In 2014, SIDS governments collectively committed to ending malnutrition by 2030. A deeper understanding of food sourcing is essential to achieve this goal.

## 1. Introduction

Small Island Developing States (SIDS) are a group of 57 countries and territories worldwide, with 37 of them being recognised as United Nations (UN) Member States (https://www.un.org/ohrlls/content/about-small-island-developing-states (accessed on 14 July 2025)). SIDS have a combined population of approximately 65 million and are located in three geographical regions, the Caribbean, the Pacific, and the Atlantic, Indian Ocean and South China Sea (AIS) (https://sdgs.un.org/topics/small-island-developing-states#list_of_sids (accessed on 14 July 2025)). As observed in the previous literature [1], SIDS are a varied group that does not fit neatly into one category. Despite their differences, the UN recognises them as an important collective due to their shared characteristics, and SIDS governments use the designation to present a unified voice and amplify their influence in addressing critical issues within the UN system.

SIDS share complex health challenges related to nutrition and food security, undergoing a rapid nutrition transition characterised by a progressive shift towards inadequate availability and access to local and healthy food [2]. This shift promotes diets that are increasingly dominated by processed imported foods, typically micronutrient-poor, and high in sugar, salt, and fats. Consequently, SIDS have some of the world’s highest rates of non-communicable diseases (NCDs) [3]. Coupled with high rates of type 2 diabetes and obesity, many SIDS also experience high levels of childhood stunting and micronutrient deficiencies, causing a substantial burden of diet-related conditions [2]. For example, eleven Caribbean SIDS have obesity rates above 30 per cent among adult women [4], while seven of the world’s top ten countries for diabetes prevalence are Pacific SIDS [5]. Child stunting rates exceed 20 per cent in several SIDS including Papua New Guinea (49.5%), Solomon Islands (32.8%), Vanuatu (28.5%), Nauru (24%), Guinea-Bissau (27.6%) and Haiti (21.9%) [4] while SIDS with a high prevalence of anaemia in women of reproductive age include Sao Tomé e Príncipe (42.7%) and Guinea-Bissau (44.6%) [4]. Although these malnutrition challenges may vary from island to island, these are issues of common concern. Some of the major factors driving these shared issues are structural constraints, climate change, profound changes in food systems associated with globalisation and market liberalisation agreements, and inadequate local agriculture systems shaped by legacies of colonialism [6].

Understanding foodscapes, the spaces that shape how people access and consume food, is crucial for addressing nutrition and health-related challenges. As a result, researchers in these fields are increasingly focusing their attention on these spaces, which are recognised as complex and context-specific [7,8]. They can encompass a wide range of food sources, be influenced by both external factors and individual preferences, and have characteristics that vary from place to place [9]. However, research specifically examining food sources in the context of SIDS remains limited. Similarly, we suggest that exploring the connections between food sourcing and food consumption is crucial for understanding dietary patterns in this context. Reflecting on the implications of these food pathways—such as acknowledging that what is grown or bought may not always be consumed by the same individual—is essential. Given the dynamic nature of foodscapes and the rapid changes associated with globalisation, studying these dynamics over time through a transitioning lens is important.

In this study, we systematically reviewed the peer-reviewed and grey literature to identify, explore and map existing food sources and their interconnections in SIDS. Our aim was to provide contextual details of these food sources, exploring their links with broader social, cultural, economic, and environmental dimensions, and focusing on their changes over time. We analysed the geographical representation and conceptualisation of the existing evidence to highlight research gaps. Finally, we discussed key elements for future research, policy, and practice on leverage points for better diets, health and food sovereignty.

The overarching research question was: What are the existing sources of food for consumption in SIDS in terms of their variety, extent, and nature?

To examine this, we set out the following set of specific research questions:where is the existing evidence drawn from?what are the links between food sources and health and nutrition, and with broader aspects such as social, cultural, economic, and environmental dimensions?what are their changes over time and across generations?what are the interconnections between these food sources?what is the extent of the existing evidence, and how has it been conceptualised?

## 2. Methods

We used a systematic scoping review method to identify a wide range of literature related to food sources in SIDS. We followed the PRISMA Extension for Scoping Reviews guidelines [10] and developed a study protocol, which was registered in OpenScience Framework (https://doi.org/10.17605/OSF.IO/7ME4R) to ensure transparency and reduce duplication of work.

Our review included both peer-reviewed and grey literature. In parallel to this study, we conducted a separate analysis about the value of adding grey literature in evidence synthesis and its contribution to equity-driven global health research. Further detailed methodological descriptions and findings concerning grey literature can be found in previous publications [11,12].

### 2.1. Search Strategy

We searched across 14 peer-reviewed databases (Medline, Embase, CINAHL, PsycInfo, Global Health, Web of Science, Scopus, Assia, Econlit, PubAg, WPRIM, LILACS, MedCarib, SciELO) and 17 grey literature repositories and websites (PAHO IRIS, WHO IRIS, FAO, AGRIS, WFP, IFAD, UNICEF, UNOPS, UNOHRLLS, Hungry Cities Partnership, Resource Centres on Urban Agriculture and Food Security (RUAF), International Institute for Environment and Development (IIED), ShareCity database, Caribbean Agribusiness, The Pacific Community (SPC), The Caribbean Public Health Agency (CARPHA), and The Conversation) to cover a wide range of academic disciplines and geographical areas.

We tailored search strategies for each data source using terms related to ‘food sources’ and ‘SIDS’ (including as search terms the names of all SIDS as listed by the UN in https://sdgs.un.org/topics/small-island-developing-states#list_of_sids (accessed 14 July 2025)). These terms were informed by earlier research in similar areas [13,14] and in consultation with key experts from the review team (see Appendix A for full search strategies). We piloted a subset of studies retrieved from initial search strategies and applied further refinements after evaluation of their relevancy (see Appendix A for refinements in eligibility criteria). We included studies published from January 1992 up to June 2021, and we did not apply restrictions in language or study type. Our data range is due to major impacts in global food systems since the World Trade Organisation (WTO) agreements [15]. We conducted all searches between 11 and 25 June 2021.

### 2.2. Screening

We used EndNote and on-screen manual review to remove duplicates in the peer-reviewed and grey literature, respectively. We used COVIDENCE (https://www.covidence.org) and Microsoft Excel for study screening of the peer-reviewed and grey literature, respectively. For both datasets, we piloted a random sample of references at each screening stage until reviewers reached a consensus agreement of at least 90 per cent on eligibility criteria. Then, two independent researchers from the review team double-screened all titles, abstracts, and full texts. Any disagreements were resolved by another member of the team acting as an arbiter.

### 2.3. Data Charting

We developed a data charting form and a list of hypothesised categories following an inductive approach. We piloted the form with a random sample of included studies and refined it accordingly. The refined form included categories covering details on study conceptualisation, design, and approach, as well as comprehensive details on food source characteristics (see Appendix A for the full charting form and guiding notes). One reviewer charted all data, with 10 per cent of the studies double charted in only selected fields by a second reviewer.

### 2.4. Analysis

We used Microsoft Excel and Datawrapper (https://www.datawrapper.de) to describe the included evidence with a series of figures illustrating the geographical expansion, the overall number of papers published by year and types of food sources assessed, and the sectors and conceptual frames involved in the studies by food source types and sub-types over time. Our eligible dataset included studies reporting both qualitative and quantitative data. Given the heterogeneity of the evidence in terms of design and methodology, we used narrative synthesis to integrate the charted data and propose our classification of food sources in SIDS. We described the nature, extent and function of each food source and their changes over time, and mapped their interconnections using the visualisation platform for mapping systems, Kumu (https://kumu.io). Changes over time were assessed by analysing the extracted data from each study, synthesising reflections provided by authors within the discussion sections of included studies, as well as any historical perspectives that offered insights into evolving trends. It is important to note that our approach did not involve assessing changes over time through longitudinal studies or comparing results across different publication years. Instead, we compiled and synthesised the authors’ reflections on trends or transitions observed for each food source.

## 3. Results

We screened 10,126 records from database searching and 318 documents from grey literature searching, of which 56 articles and 96 documents met our inclusion criteria and have been included in the analysis (Figure 1). From the 152 studies included, we yielded 198 data units, as some studies were multi-country studies.

### 3.1. Geographic Coverage of Evidence

This section addresses the first research question: where is the existing evidence drawn from?

The included evidence represented a wide geographical breadth, with studies conducted across 39 countries spanning all three SIDS geographical regions. The datasets predominantly centred around individual countries (83%), while a fraction of them adopted a broader regional perspective (13%), and a smaller number focused on cities (4%). Countries in the Pacific region (21 SIDS in total) accounted for the largest proportion of datasets (51%), followed by countries in the Caribbean region (29 SIDS; 27%). Furthermore, 6% of the datasets were drawn from countries in the AIS region, the region with the SIDS countries (nine SIDS) (Figure 2).

### 3.2. Our Proposed Classification of Food Sources Based on Evidence

This section addresses the second research question: what are the links between food sources and health and nutrition, and with broader aspects such as social, cultural, economic, and environmental dimensions?

For every dataset, we extracted contextual details of the identified food sources. These details included information on their nature, scale of use, role, the population groups relevant to them, and the foods involved (see Appendix A for full descriptions of food sources). The analysis of the data extracted, alongside building on an existing food source categorisation previously used for the Community Food and Health (CFaH) project [13], led us to develop a comprehensive classification of food sources in SIDS (Figure 3).

We reported summary descriptions for each food source in Table 1, highlighting the most common characteristics observed across the evidence reviewed. Full descriptions for each food source, including context-specific details, are provided in the Appendix A.

### 3.3. Changes in Food Sources over Time and Between Generations

This section addresses the third research question: what are their changes over time and across generations?

Evidence from Haiti, Fiji and Vanuatu described a gradual shift in food preferences and habits from reliance on local foods to increased consumption of imported rice and store-bought food items, attributed to prolonged food aid provision [16,18,20,26]. This transition was reported to lead to notable transformations in dietary patterns over time and across generations [16,17,18]. Evidence from Vanuatu attributed the extended provision of food aid, particularly emergency aid, to the lengthy periods required for local crops to recover after disasters, such as droughts, with yam yields often needing 1–2 years to reach pre-drought levels [18]. Other evidence from Vanuatu, looking at food aid, highlighted the more frequent and severe climate events over the years in this region [17], which could also explain the reported prolonged aid in this setting.

Similarly, evidence from the Caribbean and Pacific regions reported that over time, grocery sources have increasingly replaced informal grocery sources and marketplaces, influencing population consumption patterns and posing challenges for smallholder producers [31,32]. This change was reported to raise concerns about the nutritional quality and sustainability of diets [36,43,45,110]. However, evidence from communities in the Dominican Republic and Fiji reported a persisting preference for marketplaces, especially when the socio-cultural benefits of these spaces were considered, and when specific fresh products such as fresh fruit, vegetables, and certain meats were bought [35,44]. In Singapore, younger generations were shown to have a preference for the convenience and variety offered by hospitality sources such as fast-food chains from the USA and Europe, which were considered fashionable, reflecting broader cultural transitions towards fast-food consumption [68,75].

Evidence from Solomon Islands, Fiji and Vanuatu noted that shifts in economic structures, such as the move towards cash-based economies, have influenced food sourcing practices from wild or own-grown food sources to a reliance on purchased food sources, particularly grocery stores and in urban areas [18,35,114]. As seen in evidence from the Caribbean region, several Pacific countries and the Seychelles, climate change was reported to have intensified this shift; rising temperatures and water scarcity discouraged food cultivation, affecting local production [18,43,61,77,97,109,110,114,119]. Additionally, socio-political factors and globalisation were reported to have increased availability and reliance on imported rice and staples, leading to a decline in cultivating traditional crops, particularly among younger generations, as imported goods dominate local markets [18,32,37,39,43,79,80,83,102,109,110]. Urbanisation, tourism growth, and land development were reported to have further reduced available land for gardening and farming, worsening its decline [18,110]. 

Most of SIDS’ cultures were described as communal, with the sharing of work and food having a central role. Food sharing was broadly reported as a type of social safety net, ensuring community well-being in the Caribbean and Pacific regions and Singapore [16,18,26,27,28,30,33,43,44,50,58,63,80,106,111,125,134,137,143,145,146]. However, a shift from communal support towards individualism was also noticed by the evidence reviewed [18,27,28,134,137,138,166]. This shift was described as a threat to the communal use and sharing of natural resources, such as marine resources. For example, factors such as easier access to fishing grounds and markets, evolving fishing techniques, and the necessity for fishermen to earn income were reported to have contributed to this shift towards individualistic approaches in the Pacific region [134,137]. In Tonga, improvements in transportation and storage were reported to have allowed fishermen to work and access resources more independently [137]. Foreign aid projects aiming to strengthen market structures were reported to have further reduced communities’ reliance on communal support [137]. Other factors, such as social changes in family structures, were reported to have made food sharing less important [18,138], although one example from Fiji reported a resurgence of a sense of “togetherness” due to the COVID-19 pandemic [27]. Climate extremes and reliance on imported foods were reported to have altered the type of local foods shared during feasting in Guam, Fiji and Vanuatu [18,146,147].

As seen in evidence from countries in the Caribbean and Pacific regions, state food sources were reported to have undergone significant transformations over time, adapting to evolving societal needs and challenges. Described as initially aimed to address immediate hunger and poverty, these programs were reported to have evolved into more integral components of broader social safety nets, improving universal access to food, education and promoting well-being [28,41,53,56,59,60,117,148,149,150,151,152,153,154,155,156]. Despite their importance, some programs in Cuba, the Dominican Republic and Haiti were reported to lack clear legal frameworks, leading to coordination challenges with other support initiatives [148]. Furthermore, the increase in population was reported to add pressure on resources, requiring larger programme capacity to properly assist vulnerable populations in Guyana [151].

In the Pacific region, wild food sources were reported to retain their significance in rural areas, but a shift towards monetisation has led to trading them for cash. This shift was reported to result in changes in the distribution of wild foods and the involvement of intermediaries, causing competition and overexploitation [43,54,114,137,161,164]. The importance of being near wild food sources in rural communities was described as changing, reflecting the value of people’s time [18]. For example, the importance of being close to wild foods was not solely about sourcing food, but also about showing social status [18,130]. Similarly, in urban areas in Jamaica, the availability of imported food as substitutes for traditional foods was reported to lead to a decline in the collection and consumption of wild foods, resulting in social stigma associated with collecting wild foods, and higher socioeconomic status associated with the ability to buy imported food items [128]. Despite the reported decline in the collection of wild foods, adolescents and children in Solomon Islands were reported to show a strong interest in preserving traditional practices such as fishing and collecting wild foods [36], while women in Fiji were described as expanding their roles in fishing over time, targeting a wider range of habitats and a greater diversity [160]. Recently, the COVID-19 pandemic was reported to have prompted households to rely more on nearby rivers, the sea, and wild forest harvests for their food supply in these two settings [30].

### 3.4. Interconnections Between Food Sources

This section addresses the fourth research question: what are the interconnections between these food sources?

We noted the various ways in which food sources are described as overlapping and intersecting with each other. These interconnections showed the complexities of food sources and the importance of considering social, cultural, and economic factors in understanding food dynamics in SIDS. We have highlighted some examples of these interconnections and their implications in Table 2 in this article. Additionally, we have visually represented all identified interconnections in an interactive open-access evidence map: https://embed.kumu.io/1d0cb948312229153482dfebe0c7df30 (accessed 14 July 2025). Within this map, each connection between food sources is supported by excerpts from the reviewed literature, accessible by clicking on the connecting arrows.

### 3.5. Extent of the Evidence

This section addresses the fifth research question: what is the extent of the existing evidence, and how has it been conceptualised?

The number of studies reporting any measure of food source that met our eligibility criteria has increased over time, with none or 1 study published each year from 1993 to 1997, 4 studies in 2010, 10 studies in 2015, and 37 studies in 2020 (Figure 4). The increasing pattern of studies over time may reflect a growing emphasis on food issues and on SIDS following the launch of the most important UN milestones in these fields (Figure 4). Overall, the most reported food source type was *Grow* (33% of all types mentioned), followed by *Buy* (23%), *Share* (17%), and *Wild* (15%). Only 6 per cent of the reported food source types were *State* and *Aid*, each.

### 3.6. Conceptualisation of the Evidence

This section addresses the fifth research question: what is the extent of the existing evidence, and how has it been conceptualised?

The average number of different disciplines involved in each study has increased over time, with an average of 2 disciplines in 1993 compared to an average of 3.4 observed in 2021 (Figure 5). This trend suggests a growing inclination towards interdisciplinary and multisectoral research in recent years, particularly evident in peer-reviewed publications. Regarding the extent to which each discipline was represented in the studies, agriculture and land studies featured most prominently, represented in 57 per cent of the studies, followed closely by food, health, and well-being at 53 per cent. However, the latter category was more featured than the former in studies focusing on *Aid*, *Buy* and *State*. Economics and finance were involved in 32 per cent of the studies, while environmental studies and sustainability, social sciences and humanities, and community and development disciplines were represented to varying degrees at 17 per cent, 20 per cent, and 19 per cent, respectively.

We found that 43 per cent of the studies framed their research around food and nutrition security, 41 per cent on sustainable development and resilience, and 40 per cent on health and disease. Around 18 per cent of the studies centred their research within the context of diet and food environments, 15 per cent focused on equity and justice issues, and another 15 per cent on economic growth and market development. Furthermore, 13 per cent of all studies explored themes of social and cultural sustainability, and 10 per cent framed it around urban sustainability and social spaces (Figure 6).

## 4. Discussion

### 4.1. Summary of Evidence

We systematically reviewed a wide range of peer-reviewed and grey literature and identified 152 studies (198 data units) reporting food sources in SIDS (Figure 1). We described the geographical areas from which existing evidence was drawn (Figure 2) and explored the extent (Figure 4) and conceptualisation of this evidence over time (Figure 5 and Figure 6). The geographical coverage of the literature is broad, but more evidence from the Caribbean and AIS countries could support decision-making on food sources in these regions. Multidisciplinary research has increased over time, but there is a notable lack of environmental and sustainability disciplines involved in the literature, which is concerning given the context of climate change. Nearly half of the studies framed their research around food and health issues (43% and 40%, respectively), with less evidence framed around broader aspects. Future work could incorporate alternative frames to ensure a more comprehensive approach is taken to address SIDS challenges.

We proposed a classification of food sources specific for the SIDS context (Figure 3) and, for each, described the variety (types), extent (relevance), nature (characteristics) and changes over time. There is a high heterogeneity of food sources existing in SIDS that fulfil important roles beyond nutrition and are particularly important for vulnerable groups. Future work focusing on less-reported food sources such as *Share* (food exchanges), *State* (government-funded) and *Aid* (foreign-funded) is needed to fully understand their functioning while contributing to reducing inequities in SIDS. Although data from the AIS region was limited, we still found studies from this region that covered all identified food source categories, supporting the broader applicability of our classification. Future research could enrich and refine this classification as more evidence emerges, particularly from underrepresented SIDS regions. We developed an interactive open-access evidence map that outlined the interconnections between food sources following our proposed classification. There are numerous interrelations between food sources, showing that pathways from sourcing to consumption can be unexpected and complex. It is important that future work recognises these connections to better understand dietary patterns in SIDS and effectively address nutrition and health-related challenges.

### 4.2. Strengths and Limitations

We included both peer-reviewed and ‘grey literature’, defined here as not controlled by commercial publishers. Incorporating grey literature posed conceptual and practical challenges related to searching, reporting, and analysis. While we acknowledge these limitations, we aimed to search for them as thoroughly and systematically as possible to draw in additional local perspectives and build a fuller understanding of the topic. We also highlight that grey literature sources can vary widely in purpose, scope, and reporting standards, which introduces additional complexity in our analysis. This could have influenced, for example, our assessment of how the existing evidence has been conceptualised across disciplines and thematic framings (our fifth research question), as these may reflect the priorities and perspectives of the organisations producing the grey literature. Nevertheless, its inclusion added important new understanding to our review and increased the representation of local expertise. We shared the methodological details (Appendix A and [11,12]) to support future researchers aiming to incorporate this body of literature in their reviews. We included a wide range of disciplines and types of evidence, which posed challenges to the inclusion and analysis of highly heterogeneous data. To make it manageable, we iteratively refined our eligibility criteria in discussion with the review team, acknowledging the iterative nature of the systematic scoping review method [173]. While we did not critically appraise the quality of included studies, this aligns with the selected scoping method, which is intended to summarise the current landscape and guide future systematic reviews and meta-analyses. Our approach to grouping disciplines and framing of evidence into categories was guided by thematic similarities observed in the data. We provided the details of the classification systems in the Appendix A. Different research teams may have come up with different groupings in terms of number and content, which could affect the frequencies shown in the heatmaps. Finally, we ran our searches in June 2021, thus retrieving fewer studies for 2021 than in previous years (Figure 4). However, we included evidence spanning a broad period (January 1992–June 2021) and focused on using a transitioning approach for our analyses. We believe that this approach has allowed us to provide an accurate representation of the trends of food sources over time. Nevertheless, it is important to emphasise that our analysis relies on narrative synthesis of authors’ reflections rather than quantitative trend data due to the nature of the available evidence. Future studies collecting systematic, comparable quantitative data to enable statistical analyses of trends in food sources over time and across generations would be valuable.

### 4.3. Policy and Research Implications of This Study

Our study reveals important evidence gaps that might guide future work of researchers from a broad range of disciplines and sectors. The comprehensive classification and details on food sources might allow policymakers and practitioners to have a contextual understanding of the functioning and significance of these, enabling them to make informed decisions without relying on evidence derived from other contexts. For example, recognising the significance of informal grocery and marketplaces as equalisers of social status [20,44,70], or acknowledging the broad positive impact of community aid compared to emergency aid [13,26,28,29,30] may lead to more effective, equitable, and sustainable solutions in SIDS.

Applying a transition lens in our review by discussing the social, environmental, and economic shifts and how they interrelate [174] has enabled us to demonstrate that people’s interactions with food sources are dynamic, undergoing rapid changes over time and influenced by global factors. These interactions do not follow linear trends; for example, practices like food sharing through bartering have resurged during the COVID-19 pandemic [141,143]. This complexity highlights the need to move beyond simplistic dichotomies in understanding food sources and foodscapes, which is important for informing future research and policy aimed at enhancing community resilience.

Specifically, our synthesis illustrates how interactions with food sources have evolved in multiple, sometimes overlapping directions over time and across generations. We observed increasing engagement with purchased and imported foods alongside continued or renewed interest in local and own-grown sources, particularly in response to shocks like the COVID-19 pandemic. Similarly, while food sharing has in some contexts given way to more individualised practices, it remains resilient or resurges under certain circumstances. State food programs have also adapted, expanding from immediate relief to more integrated roles within social support systems. These patterns underscore the fluid and context-dependent nature of food systems transitions.

Our interactive map showing the interconnections between food sources enables broad communities of research, policy, and practice to reflect on the pathways linking food sourcing and consumption and their implications in SIDS. For example, the efforts made by many islanders to provide food for community feasts, such as giving part of the home garden production to the church, illustrate their willingness to keep both their social ties and cultural values despite facing economic and social changes that make these contributions more difficult. The map was also designed as a dynamic open resource, intended to evolve with new insights and contributions from this broad community, by including new connections or by supporting or challenging the existing ones. This interactive map also has potential utility as an accessible thinking tool for diverse non-academic audiences, including policymakers and community actors, enabling them to visualise and engage with complex evidence in an intuitive way. Our findings call for syndemic approaches [15] that recognise the diverse social, economic, political, and environmental determinants that affect health outcomes in the SIDS population to inform future work, guide research agendas, and establish funding priorities to better address SDGs in the SIDS context.

Throughout this review, we engaged with a body of literature that used different terms to refer to SIDS [175,176,177,178,179,180]. These terms included BOSS (Big Ocean Sustainable States or Big Ocean Sovereign State), BOS (Big Ocean States), or LOS (Large Ocean States) and were often used by grassroots organisations or stakeholders in the fields of ocean sustainability and climate change that emphasised the rich marine resources of certain SIDS [181]. Although there is no officially recognised list of SIDS countries that have agreed upon a new term, reconceptualising what they call themselves and changing the narrative on vulnerability and stereotypes of need has been an ongoing topic for at least a decade [182]. Although not yet reflected in most of the academic literature or within the UN system, future research, policy, and practice should acknowledge these terms to better support their alignment with the Sustainable Development Goals.

## 5. Conclusions

In this study, we have presented evidence to address the overarching research question: What are the existing sources of food for consumption in SIDS in terms of their variety, extent, and nature?

We found a high heterogeneity of food sources in SIDS that fulfil important roles beyond nutrition, especially for vulnerable groups. These food sources are undergoing significant transformations over time and across generations, with changes linked to various factors. The analysis of the evidence allowed us to develop a food source classification specific to the SIDS context, and map numerous interconnections between them. These interconnections indicated that pathways from food sourcing to consumption can be complex and unexpected.

Interestingly, nearly half of the included evidence was drawn from grey literature, which provided valuable insights and rich contextual details, especially from documents involving local authors and contributors. During analysis, we also engaged with evidence highlighting more positive and empowering terms for SIDS, along with growing debates on inequities in global health research.

## Figures and Tables

**Figure 1 nutrients-17-02353-f001:**
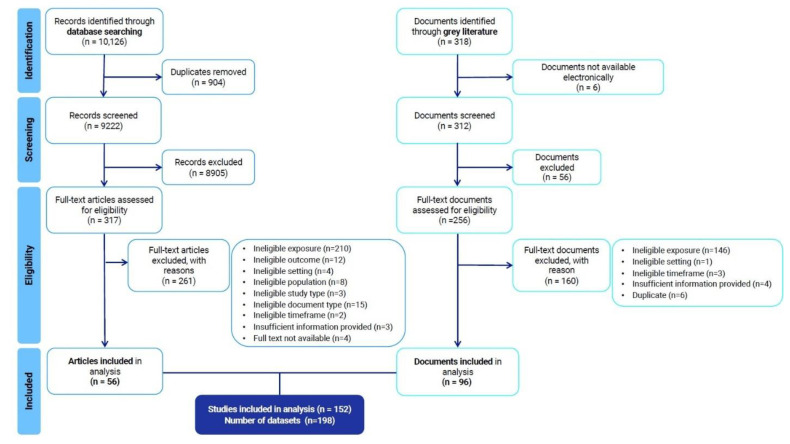
Preferred Reporting Items for Systematic Reviews and Meta-Analyses flow chart. The figure summarises the selection process of eligible studies reporting food sources in SIDS. We included studies published from January 1992 up to June 2021, with no restrictions on language or study type. Studies were excluded if they did not focus on SIDS, did not describe actual food sources (e.g., focused solely on food transport, distribution, or hypothetical scenarios), or reported food sources unrelated to the human food environment. Full search strategies, additional details on inclusion and exclusion criteria and citations of included studies have been provided in the Appendix A.

**Figure 2 nutrients-17-02353-f002:**
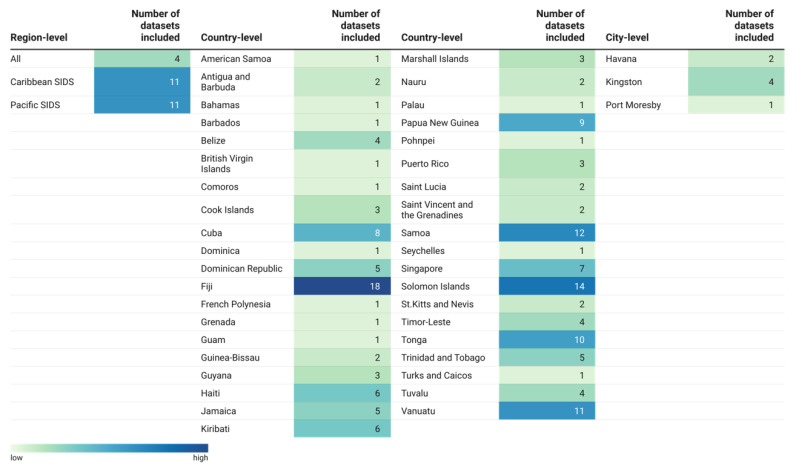
Geographical distribution of included datasets. The figure summarises the number of datasets reporting any measure of food sources by region-, country-, and city-level resulting from our systematic scoping review. The Pacific region included 13 UN-member SIDS, 7 non-UN member SIDS and 1 non-UN SIDS territory (i.e., Tokelau). The Caribbean region included 16 UN-member SIDS and 13 non-UN member SIDS. The AIS region included 8 UN-member SIDS and 1 former UN-member SIDS (i.e., Bahrain). A full list of the included countries and territories has been provided in the Appendix A.

**Figure 3 nutrients-17-02353-f003:**
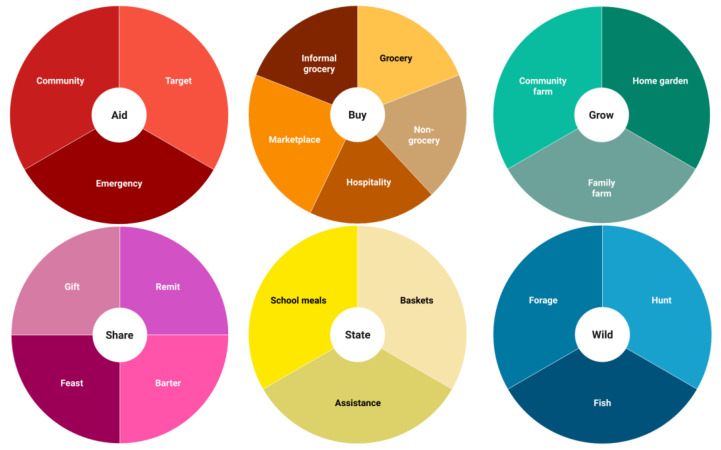
Proposed classification of identified food sources in SIDS. The figure summarises our proposed classification for food sources. We have reported summary descriptions for each food source in Table 1 in this article. Full descriptions of each food source, including context-specific details, have been provided in the Appendix A.

**Figure 4 nutrients-17-02353-f004:**
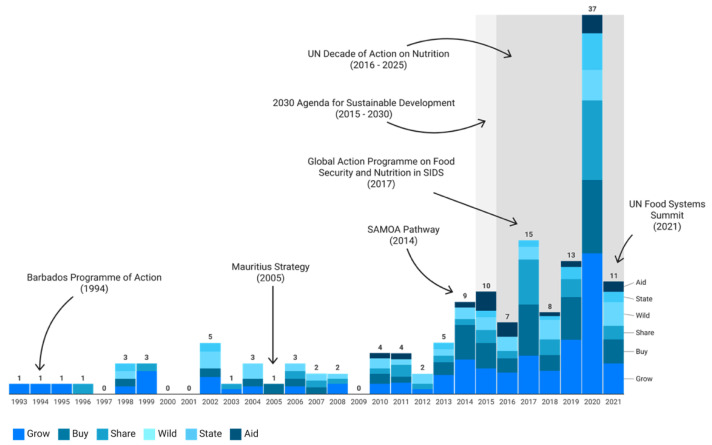
Number of studies reporting food sources in SIDS and the proportion of types over time. The figure displays the number of studies reporting any measure of food sources by year of publication from 1993 to 2021 (columns). As literature searches took place during June 2021, the data is incomplete for that year. Each year, we report the types of food sources identified following our classification (*Aid, Buy, Grow, Share, State, Wild*) and their proportion over all instances that year. The figure highlights the UN programmes of action in support of SIDS (the Barbados Programme of Action [167], the Mauritius Strategy [168], the SAMOA Pathway [169], the Global Action Programme on Food Security and Nutrition in SIDS) [2], and the most important UN milestones in food related issues (the Agenda 2030 for Sustainable Development [170], the UN Decade of Action on Nutrition [171] and the UN Food Systems Summit [172].

**Figure 5 nutrients-17-02353-f005:**
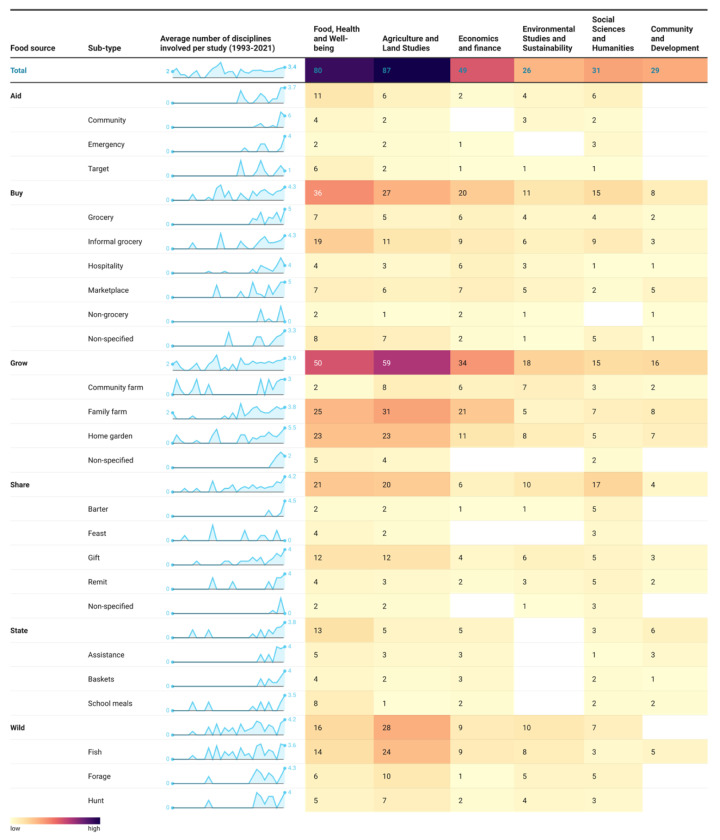
Disciplines involved in studies by food source types and sub-types over time. The heat map cross-references food source types and sub-types based on our classification to show the frequency of studies that reported disciplines by category (food, health and well-being, agriculture and land studies, economics and finance, environmental studies and sustainability, social sciences and humanities, and community and development). The figure also summarises the average number of different disciplines involved in studies over time, categorised by food source type and subtype. The details of the classification system used for disciplines are given in the Appendix A.

**Figure 6 nutrients-17-02353-f006:**
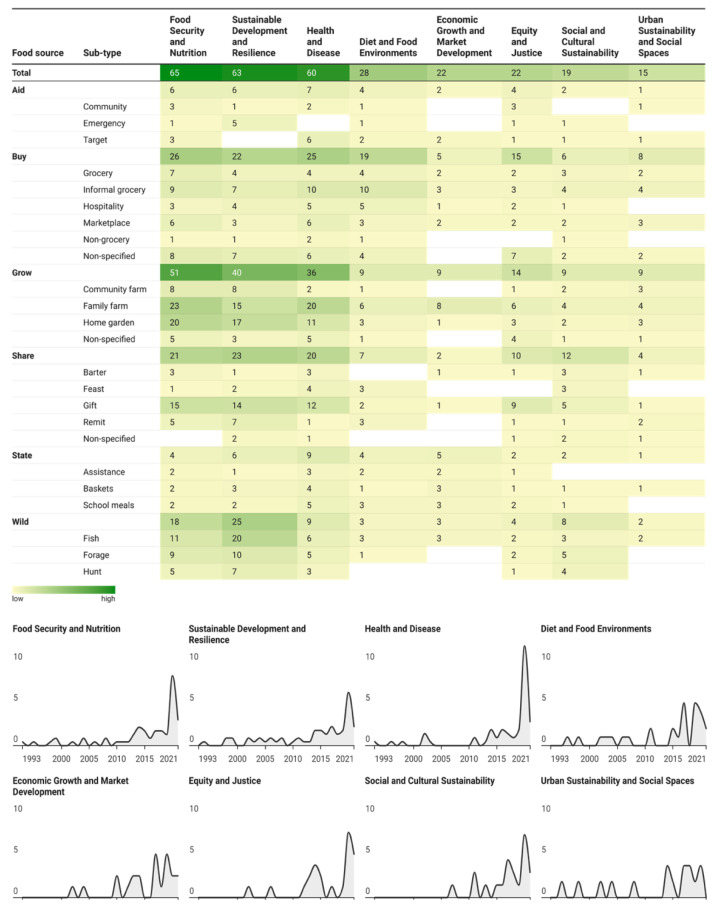
Conceptual frame of the studies by food source types and sub-types over time. The heat map cross-references food source types and sub-types based on our classification to show the frequency of studies that framed their research by theme (food security and nutrition, sustainable development and resilience, health and disease, diet and food environments, economic growth and market development, equity and justice, social and cultural sustainability, urban sustainability, and social spaces). The figure also illustrates the evolution of studies framing their research within specific theme groups over time. Each theme group is represented by a separate line chart showing the normalised number of studies over time to show the relative growth or decline of studies within each theme, independent of increases in the number of total studies. The details of the classification system used for frames are given in the Appendix A.

**Table 1 nutrients-17-02353-t001:** Summary descriptions of the food sources classification.

Sub-Type (Type)	Summary Description	Location and Studies
Emergency (Aid)	Reported as immediate short-term assistance during and after disasters, such as extreme weather events. Typically provided by foreign governments, international organisations, humanitarian agencies or philanthropic organisations. Commonly reported foods consisted of imported dry and shelf-stable products such as instant noodles, rice, flour, biscuits, and tinned meats and fish (see Appendix A for a classification of food items by food group). While it played a crucial role in alleviating hunger and increasing food access and availability of a few food items, these were reported to contradict public health messages on healthy diets [16]. Particularly impactful for populations affected by climate extremes, especially agricultural-dependent families facing significant losses due to the reliance on garden produce for subsistence.	Vanuatu [16,17,18,19], Fiji [20].
Target (Aid)	Described as food aid provided by foreign donors that additionally supported existing national programmes and local initiatives for a longer period compared to *Emergency*. Reported foods included a mix of local and imported items like cereals, pulses, nuts, and vegetable seedlings, serving as a safety net addressing short-term hunger, increasing household food income, children’s school attendance, and access to nutrient-rich food. Despite this, it was reported to lack food diversity and not meet the food energy requirements [21]. Commonly reported as relevant for individuals suffering from severe malnutrition and disease, including food-insecure primary school children. However, it was reported to struggle to reach the most disadvantaged children from the poorest and most vulnerable communities [22].	Pacific region [23], Cuba [24], Dominican Republic [24], Haiti [21,24,25,26], Guinea-Bissau [22], Fiji [27].
Community (Aid)	Reported as food aid funded by locally based non-governmental organisations, charities, social enterprises, churches, and voluntary groups. Often relied on volunteer donations and included initiatives such as providing meals during school holidays to children in need, and health awareness activities [28]. This type of aid was reported to provide some fresh meat and vegetables in addition to food staples. It played diverse roles, from improving food security to mental well-being, nutrition and health education. It was especially important for those individuals facing food insecurity [26], and addressing feelings of loneliness and social isolation across various income levels [29].	Singapore [29], Fiji [13,30], Solomon Islands [30], Haiti [26], Belize [28], Saint Vincent and the Grenadines [13].
Grocery (Buy)	Referred to by the evidence as ‘hypermarket’, ‘supermarket’, ‘big-box style supermarket’, ‘modern convenience store’ and ‘Western-style grocery store’. Described as modern stores, often owned by multinational corporations. These establishments, supplied by wholesalers, were reported to offer a wider range of food items, albeit at higher prices [31,32]. While grocery stores were described as providing convenience and bulk purchasing options, they were visited less frequently, possibly due to the practice of monthly bulk purchases influenced by discount offers [33,34]. Positively associated with higher reported intakes of fruit and vegetables and greater dietary diversity, as well as increased consumption of sugar-sweetened beverages, red and processed meats [13]. Described as the preferred choice for purchasing processed and manufactured food products [35]. They were reported to symbolise status and modernity, offering imported foods associated with affluence and convenience [36,37]. Especially relevant for high-income households.	Caribbean region [31,38], Solomon Islands [30,36,39], Pacific region [32], Kingston (Jamaica) [33,34], Jamaica [40], Fiji [30,35,41], French Polynesia [37], Samoa [42], Tuvalu [43], Dominican Republic [44], Turks and Caicos [45], Trinidad [46], Saint Vincent and the Grenadines [13].
Informal grocery (Buy)	Referred to by the evidence as ‘street-food vendor’, ‘corner shop’, ‘food truck’, ‘colmado’ (term used in evidence from the Dominican Republic which refers to a mix between a small supermarket and a bar), ‘small shop’, ‘roadside stationery’, and ‘itinerant stall’. Generally reported as having a decentralised structure with various suppliers, transporters, and vendors, predominantly comprised of small, family-owned businesses with minimal capital investment. Despite potential limitations in stock and pricing, these establishments were reported to offer proximity, monthly credit facilities, and convenience [35,44,47,48], often concentrating in commercial hubs and adapting to consumer needs with flexibility [34,49,50]. Vendors were described as adapting to the socio-economic conditions of their clientele by selling food in smaller portions, thus making goods more affordable. Also, they adopted customer-friendly practices to cultivate loyalty by including extra portions at no charge or by providing credit [33,44,51]. This food source was reported to play an important role in ensuring food security for poor urban households and remote rural villages without access to transportation [42,52], while also serving as a source of income for women vendors [23,53]. Described as offering a wide range of foods from local fresh produce to bulk products such as rice and beans, to packaged snacks, soft drinks, sauces, and canned processed foods.	Pacific region [23,54], Caribbean region [31,53], Kiribati [55], Samoa [42,55], Solomon Islands [55], Vanuatu [55,56], Pohnpei [57], Barbados [52], Timor-Leste [58], Kingston (Jamaica) [33,34,51], Jamaica [40,59], Cuba [60], Fiji [35,41,61], Havana (Cuba) [48], Belize [28], Haiti [49], French Polynesia [37], Tonga [50,55], Dominican Republic [30,44,47], Turks and Caicos [45], Trinidad [46].
Non-grocery (Buy)	Referred to by the evidence as ‘pharmacy’, ‘bookstore’, ‘gas station’, ‘internet retail’, and ‘door-to-door retail’. It was reported to offer limited selections of ultra-processed foods and drink products, with online retail playing a minor role, and primarily used during COVID by the at-risk population to avoid crowds in stores [31,62,63].	Caribbean region [31,62], Singapore [63].
Marketplace (Buy)	Referred to by the evidence as ‘central market’, ‘domestic market’, ‘local market’, ‘municipal market’, ‘town market’, ‘city market’, ‘formal market’, ‘wet market’, ‘fish market’ and ‘farmer market’. Marketplaces were reported to feature stalls with various fresh produce and spices [33,41,64]. Despite challenges like poor sanitation and vendor vulnerability [34,65,66,67], these markets were described as perceived providers of fresher and healthier options due to the presentation of unpackaged food and the perceived knowledge and personal attention of vendors [34,35,68,69]. They were described as places where people felt less judged by appearances compared to grocery, serving as social equalisers [68,70], and to provide stable gathering points where people connected to their culture and traditions and found a sense of familiarity in the face of rapid societal shifts, encouraging social interactions and trust among diverse groups beyond family ties [68]. They were reported to be particularly important for women, lower-income earners, and the elderly [54].	Papua New Guinea [65,66,67], Port Moresby [67], Barbados [69], Kiribati [55], Samoa [55], Solomon Islands [30,55,64,71], Tonga [55], Vanuatu [55], Pacific region [54], Belize [72], Kingston (Jamaica) [33,34], Cuba [60], Fiji [30,41], Haiti [26,70], Singapore [68], Turks and Caicos [45].
Hospitality (Buy)	Referred by the evidence as ‘hotel’, ‘restaurant’, ‘kava bar’(only in evidence from the Pacific region to refer to bars specialised in serving drinks made of *kava*, a native plant), ‘hawker centre’ (term used in evidence from Singapore to refer to open-air food complexes with multiple vendors selling local prepared foods), ‘food court’ (found in larger shopping malls, an air-conditioned version of hawker centres), ‘takeaway’, ‘coffee shop stall’, and ‘corporate fast-food chain’. They were reported to offer very diverse food options and environments, with quick, affordable meals and extended menus (except for kava bars) [31,33,41,54,64,73,74]. Convenience and affordability were the main reasons reported for choosing this type of food source. The rise of Western fast-food chains was reported to raise health concerns due to associations with obesity and chronic diseases [74]. Younger individuals, particularly students, were reported to be the main population group affected by fast-food chains [13], as they frequented these places for socialising and studying, seeking a sense of belonging and identity within peer groups [73]. Instead, the elderly found their socialising spaces in hawker centres [73,74,75]. In Fiji, kava bars and street stalls were described as particularly relevant for urban dwellers to purchase cooked seafood and finfish [13,41].	Dominican Republic [31], Caribbean region [38], Samoa [55], Pacific region [54], Kingston (Jamaica) [33], Fiji [13,41], Singapore [73,74,75], Solomon Islands [64], Saint Vincent and the Grenadines [13], Trinidad [46].
Home-garden (Grow)	Home garden included the terms ‘urban garden’, ‘village garden’, ‘backyard garden’, ‘kitchen garden’, ‘domestic garden’, and ‘sup-sup garden’ (term used in evidence from Solomon Islands, which refers to organic backyard gardening). They are typically located adjacent to residences or nearby areas. These gardens varied in size but were often small, included recycled containers for cultivation, and followed organic agricultural practices [60,76,77,78]. Growers often lacked formal training [79], and primarily cultivated crops for household consumption [28,36,78,80,81], exchange, and gifting [82,83], with limited commercial sales [17,19,30,48,79,84,85,86,87]. The main foods cultivated included vegetables, fruits, herbs and crops easily adaptable to urban spaces, alongside occasional livestock (mainly chickens). Generally reported as enhancing food security, nutrition, and household economies by providing fresh produce, supplementing diets, and generating income, while also promoting social well-being through community ties, empowerment, and cultural preservation. Often described as a food source that offers resilience against economic downturns, environmental challenges, and stress, particularly evident during crises like COVID-19 (see Appendix A for context-specific insights).	Pacific region [23,76,86], Marshall Islands [85,88], SIDS regions [81], Vanuatu [16,19,56,89], Fiji [30,56,61,82,83], Samoa [56], Solomon Islands [30,36,39,56], Tonga [56,82], Tuvalu [56], Trinidad and Tobago [79,90], Jamaica [84,91], Bahamas [92], Grenada [93], British Virgin Islands [94], Timor-Leste [58,80], Kingston (Jamaica) [33,95], Cuba [60,78], Antigua and Barbuda [48,87], Belize [28], Haiti [26], Puerto Rico [96], Kiribati [82], Nauru [82], Papua New Guinea [82], Turks and Caicos [45].
Family farm (Grow)	Described as managed and operated by a family, predominantly relying on family labour. Compared to home gardens, these farms typically involved larger-scale production, a wider variety of crops, and the inclusion of diverse livestock and cash crops. They were described as using mixed cropping techniques and minimal inputs [20,97,98,99], aiming for self-sufficiency, sale and gifting, supporting both local food supply [25,100,101,102,103], and cultural obligations [43,104]. The production from family farming was described as ensuring food access to families during periods of supply shortage due to the COVID-19 pandemic [42,52,72,90,102,105,106], providing resilience against external shocks, whether economic (price spikes, global recession) or natural (cyclones, floods, droughts, pests, and diseases) [17,103,104].	Caribbean region [25,53,69,97,104], Pacific region [86,103,107,108], SIDS region [100], Marshall Islands [109], Cook Islands [110,111], Solomon Islands [71,111,112,113,114], Tuvalu [43,115], Kiribati [115], Fiji [20,27,41,111,112,116], Timor-Leste [58,117], Samoa [42,101,111,116], Nauru [118], Comoros [99], Tonga [111,113], Vanuatu [17,18,111,112,119], Papua New Guinea [111,120], Saint Kitts and Nevis [105], Saint Lucia [106], Guyana [121], Belize [72], Barbados [52], Dominica [102], Guinea-Bissau [122], Seychelles [77], Cuba [98,123].
Community farm (Grow)	Described as located on idle public lands in various urban and rural settings [123,124,125], and organised by charities, non-profit associations, cooperatives, social enterprises, or groups of neighbours. Cultivation practices were reported to occur in raised beds or existing soil with minimal external inputs. Community farms also grew a variety of crops, including starchy foods, vegetables, and fruits, alongside herbs and medicinal plants, with occasional integration of poultry and pigs. Produce was reported to be typically divided among farmers [79,124], and sold at local markets or through food box schemes to support garden operations contributing to local food supply [60,79,125,126,127], and to the tourism sector [127]. Described as educational and cultural hubs for the community [63,123,124,128], serving a socialising role [29,123,124].	Caribbean region [25], Nauru [82,118], Havana (Cuba) [129], Cuba [60,78,123,125,127], Singapore [29,63,126], Trinidad and Tobago [79], Puerto Rico [124], Papua New Guinea [82], Fiji [82], Tonga [82], Kiribati [82], Jamaica [128].
Gift (Share)	Reported as the exchange of food items within various social contexts, including schools, families, neighbourhoods, and communities [13,28,33,50,58,110]. Described as traditions of communal support, such as sharing farming produce and fish catches, especially during times of need [18,30,54,58,106,130,131]. This practice was sometimes reported to be facilitated through community initiatives promoting knowledge sharing and solidarity [29,63,132]. Food gifts to those in need were driven by moral responsibility and reciprocity [44,80,131]. Overall, food gifting reinforced social bonds and mutual assistance, serving as a form of social capital within communities.	Samoa [111,133], Pacific region [54,130,134], Cook Islands [111], Fiji [27,30,111], Papua New Guinea [111], Solomon Islands [30,111,135], Tonga [50,111,136,137], Vanuatu [18,111,138], Saint Lucia [106], Timor-Leste [58,80], Kingston (Jamaica) [33], Singapore [29,63,132], Kiribati [139], Tuvalu [140], Belize [28], Haiti [26], Cuba [125], Dominican Republic [44].
Barter (Share)	Described as the trade of food for goods or services without money [30,39,88,109,131]. It occurred between communities and saw a resurgence during the COVID-19 pandemic, facilitated by social media platforms [30,141,142,143]. This strategy proved valuable during times of uncertainty, allowing people to meet their basic needs [44,138,141].	Solomon Islands [30,39], Fiji [30,141], Samoa [142,143], Puerto Rico [131], Vanuatu [138], Dominican Republic [44], Jamaica [128].
Remit (Share)	Typically involved the exchange of food or cash between different locations, such as from a main island to outer islands within the same country, between different countries, or across diverse community settings like coastal and highland areas. This exchange often occurred within familial relationships, distinguishing it from barter, which may have lacked this personal connection. Unlike bartering, food remittances held emotional significance, as foods were believed to have unique flavours and carried strong sentimental value.	New Caledonia [88], Marshall Islands [109], Jamaica [91], Kingston (Jamaica) [33,95], Tuvalu [43], Fiji [61], Vanuatu [16,18].
Feast (Share)	Described as a common tradition in ceremonies and celebrations that was important for communities [61,88,144,145,146,147]. Sundays were reported as special “feasting days” when food was shared in a structured way, reflecting social status [61,147]. According to the evidence, feasting went beyond immediate family, lasting several days and needing careful planning [145]. During various community events, like church gatherings and holidays, feasting involved everyone contributing dishes, creating a sense of unity and responsibility. It was reported to involve large quantities of diverse foods.	Cook Islands [88], Papua New Guinea [144], Fiji [61,147], Vanuatu [145], Guam [146].
School meals (State)	Described as covering various programs, including universal schemes providing free daily meals to students in public or semi-public schools. Meals were reported to face challenges in meeting national guidelines due to funding limitations [148]. Some reported programmes consisted of preparing cooked meals in school canteens, while others prepared the lunches centrally and distributed them to schools [28,59]. Some programs were reported to integrate school gardens to enhance food diversity and freshness [41,149]. During the COVID-19 pandemic, adaptations such as take-home rations and food vouchers were reported to be introduced [150].	Caribbean region [53], Dominica [150], Guyana [150,151], Jamaica [59,150], Saint Lucia [150], Trinidad and Tobago [150,152], Aruba [150], Sint Maarten [150], Cuba [148], Dominican Republic [148], Haiti [148], Fiji [41], Belize [28,149,150].
Baskets (State)	Described as provided by the government and typically containing locally grown foods as well as rice and other staples [117]. These baskets were reported to be distributed through beneficiary cards or ration cards, offering discounts or fixed-price options [60,153]. However, consistent support was a challenge for some programmes, with only small monthly food baskets available to families in need [28,37].	Timor-Leste [117], Vanuatu [56], Belize [28,153], Cuba [60,148], Dominican Republic [148], Haiti [148], French Polynesia [37].
Assistance (State)	Described as provided by the governments in response to specific events or situations. For example, it was reported that to address the limited market access for farmers, some governments established connections between farmers and schools, allowing farmer groups to supply the feeding programs [117,148,154,155]. These programmes were reported as having introduced new and improved farming opportunities and technologies for the local market [155].	Palau [156], Marshall Islands [156], Samoa [156], Solomon Islands [154], Cuba [148], Dominican Republic [148], Haiti [148], Trinidad and Tobago [155], Saint Kitts and Nevis [155], Guyana [155], Saint Lucia [155], Belize [28], Timor-Leste [117].
Forage (Wild)	According to evidence, it involved the gathering of wild foods from natural environments such as forests, bushlands, and uncultivated farmlands. Some collectors were reported to specialise in certain foods while others gathered a variety of them [128]. Consumption of foraged foods was reported to vary with the seasons, with methods like drying tubers for storage during the wet season [157]. Foraging was reported to include diverse food types like tree crops, wild plants, fruits, nuts, staples, and root vegetables, ensuring a varied diet.	Caribbean region [158], SIDS region [159], Solomon Islands [30,36,39,114], Papua New Guinea [126], Fiji [27,30,41,83], Timor-Leste [80,157], Vanuatu [18,145], Jamaica [128].
Hunt (Wild)	Reported as typically involving individuals or small groups using trained dogs, spears, or bows and arrows [27,39]. According to the evidence, the most targeted animal was the wild pig. Other reported hunted foods included mammals, rodents, reptiles, birds, insects, and worms, as well as animal products such as honey, eggs, and birds’ nests.	Caribbean region [158], SIDS region [159], Solomon Islands [36,39,114], Timor-Leste [58,157], Papua New Guinea [126], Fiji [27], Vanuatu [145], Jamaica [128].
Fish (Wild)	Described as including open ocean, mangroves, mudflats, coral reefs, and rivers. Coastal and inland communities were reported to rely on these diverse ecosystems for animal protein. Artisanal methods like manual collection, spearfishing and hook and line fishing, or the use of small boats were common [54,134]. Gender dynamics played a significant role in fishing practices, with fishing activities often segregated based on location or habitat [20,39,160,161]. It was reported that fishing ability could symbolise social status, with individuals gaining prestige by catching more fish and enhancing their reputation within the community by sharing greater amounts of the catch [130].	Caribbean region [158,162,163], SIDS region [81], Pacific region [54,86,103,107,130,134,164], Cook Islands [110], Fiji [20,30,41,112,160,165], Vanuatu [112], Tonga [137,165], Samoa [42,165], Tuvalu [43,165], Belize [72], Dominica [102], British Virgin Islands [94], Timor-Leste [58], Papua New Guinea [126], French Polynesia [37], Kiribati [161], Solomon Islands [30,36,39,112,165].

**Table 2 nutrients-17-02353-t002:** Examples of interconnections between food sources and their implications.

Rural-Urban Food Exchanges	Implications
We saw from the evidence that food transfers of wild, grown, or bought foods occurred between relatives living in rural areas and those in urban centres. For example, in Fiji, residents of remote coastal areas in outer islands often relied on family members in urban centres to obtain fresh produce unavailable locally, facilitated by weekly boat shipments [61]. Similarly, local foods that arrived in Funafuti (the capital of Tuvalu) from the outer islands were shared extensively between neighbours and relatives [43]. In the Republic of Marshall Islands, locally grown food crops were traditionally not sold but shared or exchanged between communities [109].	These connections between rural-urban food sources have some implications. First, that wild, grown, or bought food by an individual or family is not always consumed by them; instead, food can be sent to relatives elsewhere or shared extensively with others. Second, they show that fresh produce also flows from urban to rural settings. While rural areas may be perceived as the primary source of food production, these connections highlight how urban settings can also provide fresh produce to remote rural areas.
In Tuvalu, there were exchanges of local atoll foods such as fresh and dried fish, *pulaka* (a local root rich in carbohydrates), bananas, coconut, and breadfruit for imported items such as ultra-processed foods and non-food items such as floor coverings [43]. Coastal-mountain food transfers, as seen in New Caledonia and Vanuatu, involved the exchange of wild or grown foods among family members, contributing to dietary diversity [16,88]. In Jamaica, monthly food transfers from rural to urban areas included home-grown fruits, vegetables, provisions, and meats, with urban dwellers also benefiting from occasional food and cash remittances from relatives overseas [33,91,95].	These connections show that rural-urban food transfers can involve the exchange of foods from different food groups and origins than the initially sourced (e.g., fresh fish for soft drinks, and local food crops for imported food items), or even for non-food items such as home decorations. The connections also indicate that an individual’s food consumption is not necessarily reliant on the food sources available at their place of residence. Instead, external factors such as remittances influence food access and availability. This suggests that migration patterns and remittance practices can shape food consumption and dietary patterns within communities in unexpected ways.
**Informal grocery interconnections**	**Implications**
We found from the evidence that food purchased from informal grocery outlets (including street-food vendors, corner shops, food trucks, *colmados*, small shops, roadside stalls, or itinerant stalls) was interconnected with many other food sources. For example, fisherwomen in rural areas of many Pacific islands used part of their catches for sale. Although much of the fish catch (mainly invertebrates such as bivalve molluscs, crustaceans, sea cucumber, and algae) was consumed at home or shared with relatives, a small portion was sold, mainly through informal roadside markets [54]. These women also added value to their catches, preparing traditional dishes like puddings covered in coconut cream for sale to roadside buyers [20,54,160]. Since little cash was involved, these practices were seen by policymakers and donors as less important food sources than commercial fisheries [54].	This connection between fishing and informal grocery demonstrates that fish caught is not only sold in formal markets; instead, informal roadside markets are also important in local food distribution and serve as a platform for the multifaceted roles of women in local food economies.
Similarly, an example from Jamaica showed that women who were street-food vendors opted to raise chickens at home instead of growing vegetables, after recognising a market demand for chicken sales [84].	This connection between home gardens and informal grocery sources, such as street vendors, shows that home gardens not just supply vegetables for personal consumption; instead, they show that individuals adapt their home food production based on informal market opportunities.
An example from Samoa showed that during the COVID-19 pandemic, there was a rise in families selling their surplus fruits and vegetables in roadside stalls near their homes (although with a decrease during lockdowns), which provided nearby communities with access to fresh produce [42].	This connection between family farming and informal grocery shows that during crises, food security is not solely ensured by external food aid; instead, it shows a contribution of family farming to the resilience of local food systems in remote areas during crisis.

## Data Availability

All source data generated or analysed during this study are available in the Appendix A of this article.

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
