# Peer review of "Examining Food Sources and Their Interconnections over Time in Small Island Developing States: A Systematic Scoping Review"

_nutrients, 2025, doi:10.3390/nu17142353_

Round 1
Reviewer 1 Report
Comments and Suggestions for Authors
In the curent study, the authors systematically reviewed peer-reviewed and grey literature to identify, explore and map existing food sources and their interconnections in Small Island Developing States. They provided contextual details of the food sources, explored their links with social, cultural, economic, and environmental dimensions, and on their changes over time. They analyzed the geographical representation and conceptualization of the existing evidence to highlight research gaps and discussed key elements for future research, policy, and practice on leverage points for better diets, health and food sovereignty.
Some comments:
1.Do you think grey literature provides conclusive information?
2. Point 2.1. You included studies published from 1992 up to 25th June 2021? Please clarify.
3. Figure 1 - To make the article easier to read (without consulting the supplementary material), please add a phrase regarding the exclusion criteria.
4.At discussion section please underline what changes did you found over time and across generations.
Author Response
Dear Reviewer 1,
Thank you very much for your time reviewing our paper and for the minor revisions you suggested. Please, find below our point-by-point responses to your comments.
Comment 1: Do you think grey literature provides conclusive information?
Response 1: Thank you for this thoughtful question. We do not claim that the grey literature included provides conclusive information, and we have already acknowledged these limitations in the earlier version of the manuscript (Section 4.3). Our aim was to take a broad view of eligible evidence to draw in a wide range of data sources and gain a fuller understanding of the topic. In line with this, our search strategy was broad, inclusive, and thorough, and we sought grey literature systematically to incorporate additional local perspectives missing from peer-reviewed publications.
We also recognise that including grey literature comes with conceptual and practical challenges, which we have explored in two related works: a conference abstract (Reference 11, lines 1281–1282) and a manuscript currently under review (Reference 12, lines 1283–1284). These two related works detail the methodological adaptations we made to include grey literature rigorously and transparently in this study (also provided in Supplementary Materials).
To make this clearer, we have now added a brief statement in the manuscript highlighting both these considerations and the unique value that grey literature brings to our synthesis (lines 1148–1151). We hope this addresses your comment.
Comment 2: Point 2.1. You included studies published from 1992 up to 25th June 2021? Please clarify.
Response 2: Thank you for pointing this out. We have now clarified in the manuscript that we included studies published from January 1992 up to June 2021, aligning with the dates of our search (conducted between 11–25th June 2021). This information has been revised in lines 119-120 and in line 1167 of the revised manuscript.
Comment 3: Figure 1 - To make the article easier to read (without consulting the supplementary material), please add a phrase regarding the exclusion criteria.
Response 3: Thank you for this helpful suggestion. We have now added a brief description of the exclusion criteria directly to the legend of Figure 1 (lines 175–178) to make the article easier to read without consulting the Supplementary Materials. We hope this addresses your comment.
Comment 4: At discussion section please underline what changes did you found over time and across generations.
Response 4: Thank you for raising this point. While we had already discussed changes over time and across generations in the original discussion (lines 1194-1201), we have now made these more explicit (lines 1202-1210) to ensure this is clearly highlighted and fully addresses your comment.
Thank you again for taking the time to review our paper and for helping us to improve it. We hope we have successfully addressed your suggestions in this revised version.

Reviewer 2 Report
Comments and Suggestions for Authors
I have reviewed the manuscript titled "Examining Food Sources and Their Interconnections Over Time in Small Island Developing States: A Systematic Scoping Review" by Anna Brugulat-Panés and co-authors. The manuscript presents a systematic scoping review, based on both peer-reviewed and grey literature, examining the diversity and interconnections of food sources in Small Island Developing States (SIDS). It uses a novel classification of food sources (Aid, Buy, Grow, Share, State, and Wild) and develops an interactive evidence map to visually represent these interrelations, while also analyzes how food sourcing patterns have evolved over time and across geographies.
The current study is one of the first to comprehensively map and classify food sources across SIDS, incorporating grey literature to ensure representation of local contexts. The creation of an open-access, interactive map is an innovative and useful tool for policy and community stakeholders.
Some comments for the authors are given below.
- The analysis of change over time relies largely on authors’ reflections and qualitative synthesis. To make the findings more reliable and objective, quantitative data should be used, and statistical methods for trend analysis could be applied to clearly show how things evolve over time.
- With regard to the grey literature, briefly assess the limitations of grey sources used and how they might affect interpretation.
- The evidence map is a valuable product—I suggest integrating examples of its utility more prominently in the main text.
- The AIS region is underrepresented, in comparison to Pacific and Caribbean regions. This compromises the generalizability of conclusions regarding all SIDS.
Author Response
Dear Reviewer 2,
Thank you very much for your time reviewing our paper and for the minor revisions you suggested. Please, find below our point-by-point responses to your comments.
Comment 1: The analysis of change over time relies largely on authors’ reflections and qualitative synthesis. To make the findings more reliable and objective, quantitative data should be used, and statistical methods for trend analysis could be applied to clearly show how things evolve over time.
Response 1: Thank you for this valuable suggestion. We agree that quantitative time series analyses could provide important insights and represent a promising direction for future research. However, this was beyond the scope of our review, which was specifically designed as a qualitative synthesis of the existing literature. This methodological choice was already described in the previous version of our manuscript (lines 162–165). Furthermore, trend analyses were not feasible, as the studies we included did not provide comparable raw data or consistently reported indicators required for statistical trend analysis. Available data often consisted only of narrative reflections or aggregated statistics.
To address your comment, we have now clarified our methodological scope more explicitly under ‘Strengths and Limitations’ and highlighted the importance of future studies that collect systematic, comparable data to enable quantitative assessments of trends over time and across generations (lines 1179–1183). We hope these revisions adequately address your concern.
Comment 2: With regard to the grey literature, briefly assess the limitations of grey sources used and how they might affect interpretation.
Response 2: Thank you for this helpful comment. We agree it is important to consider the limitations of the grey literature included and how these might influence our interpretation. We have further expanded on these limitations in the manuscript (lines 1148–1151), also in response to Reviewer 1’s Comment 1. Furthermore, we now explicitly acknowledge that grey literature sources can vary widely in purpose, scope, and reporting standards, which introduced additional complexity into our analysis. This complexity could have influenced, for example, our assessment of how the existing evidence has been conceptualised across disciplines and thematic framings (our fifth research question), as these may reflect the priorities and perspectives of the organisations producing the grey literature (lines 1153-1157).
Comment 3: The evidence map is a valuable product—I suggest integrating examples of its utility more prominently in the main text.
Response 3: Thank you for your positive feedback regarding the evidence map. We appreciate your suggestion to highlight its utility more prominently. We believe Table 2 already presents several examples of the interconnections between food sources and their implications, illustrating its practical value. Additionally, in the ‘Policy and Research Implications’ section (lines 1211–1219), we discuss how this tool may support policymakers and practitioners, providing further examples of its application. To further address your comment, we have now added an additional example of the map’s potential use (line 1219-1222) which is its role as a thinking tool for a wide range of non-academic partners, including policy makers and community actors, since it provides an accessible way to visualise and engage with the existing evidence.
Comment 4: The AIS region is underrepresented, in comparison to Pacific and Caribbean regions. This compromises the generalizability of conclusions regarding all SIDS.
Response 4: Thank you for this comment. We agree that the AIS region is underrepresented in our dataset, reflecting a broader gap in the available literature. As noted in our results section (lines 189–191), only “6% of the datasets were drawn from countries in the AIS region” highlighting the limited number of studies from this area that met our inclusion criteria. We also emphasised this point in our discussion (lines 1112–1114), recommending that “more evidence from the Caribbean and AIS countries could support decision-making on food sources in these regions.”. However, it is noteworthy that despite fewer studies from this region, we still identified evidence spanning all the diferent food source categories in our analysis. We therefore believe that the classifications and maps we present offer a meaningful foundation, which can be further refined and strengthened as more studies from underrepresented regions become available. We have now added a statement to the main text to make this explicit (lines 1127–1131).
Thank you again for taking the time to review our paper and for helping us to improve it. We hope we have successfully addressed your suggestions in this revised version.

Round 2
Reviewer 2 Report
Comments and Suggestions for Authors
I have reviewed the authors' point-by-point responses and the changes made to the manuscript. I am satisfied that they have adequately addressed all of my comments and suggestions.